# Auxotrophic *Lactobacillus* Expressing Porcine Rotavirus VP4 Constructed Using CRISPR-Cas9D10A System Induces Effective Immunity in Mice

**DOI:** 10.3390/vaccines10091510

**Published:** 2022-09-09

**Authors:** Hailin Zhang, Haiyuan Zhao, Yuliang Zhao, Ling Sui, Fengsai Li, Huijun Zhang, Jiaxuan Li, Yanping Jiang, Wen Cui, Guojie Ding, Han Zhou, Li Wang, Xinyuan Qiao, Lijie Tang, Xiaona Wang, Yijing Li

**Affiliations:** 1College of Veterinary Medicine, Northeast Agricultural University, Harbin 150030, China; 2Jiangsu Hanswine Food Co., Ltd., Ma’anshan 243000, China; 3Heilongjiang Key Laboratory for Animal Disease Control and Pharmaceutical Development, Harbin 150030, China; 4Harbin Vikeses Biological Technology Co., Ltd., Harbin 150030, China

**Keywords:** *Lactobacillus*, CRISPR/Cas9D10A, nutritional deficient, porcine rotaviruses, oral immunization

## Abstract

Porcine rotavirus (PoRV) mainly causes acute diarrhea in piglets under eight weeks of age and has potentially high morbidity and mortality rates. As vaccine carriers for oral immunization, lactic acid bacteria (LAB) are an ideal strategy for blocking PoRV infections. However, the difficulty in knocking out specific genes, inserting foreign genes, and the residues of antibiotic selection markers are major challenges for the oral vaccination of LAB. In this study, the target gene, alanine racemase (*alr*), in the genome of *Lactobacillus casei* strain W56 (*L. casei W56*) was knocked out to construct an auxotrophic *L. casei* strain (*L. casei Δalr W56*) using the CRISPR-Cas9D10A gene editing system. A recombinant strain (pPG-alr-VP4/*Δalr W56*) was constructed using an electrotransformed complementary plasmid. Expression of the alr-VP4 fusion protein from pPG-alr-VP4/*Δalr W56* was detected using Western blotting. Mice orally immunized with pPG-alr-VP4/*Δalr W56* exhibited high levels of serum IgG and mucosal secretory immunoglobulin A (SIgA), which exhibited neutralizing effects against PoRV. Cytokines levels in serum detected using ELISA, indicated that the recombinant strain induced an immune response dominated by Th2 cells. Our data suggest that pPG-alr-VP4/*Δalr W56*, an antibiotic-resistance-free LAB, provides a safer vaccine strategy against PoRV infection.

## 1. Introduction

Porcine rotavirus (PoRV) causes diarrhea in piglets, leading to substantial economic losses in pig farming [1]. The lesions caused by rotavirus (RV) infection are mainly confined to the digestive tract. For example, the stomach wall becomes lax and filled with curds and milk, and the small intestinal mucosa is striped or diffusely bleeds [2]. After the RV enters the intestine through the digestive tract, it replicates in the cytoplasm of the epithelial cells of the small intestinal villi. PoRV can cause dissolution and shedding of small intestinal villi cells, changes in cell membrane permeability, and damage to the structure and function of the mucosal barrier [1]. VP4 is an outer capsid protein of PoRV responsible for receptor attachment during infection. VP4 can be cleaved by trypsin into two subunits, VP5 and VP8 (located near the amino terminal), that are related to the adsorption of virus particles on the surface of host cells and invading virus cells, respectively [3,4]. VP4 protein is also a potential vaccine immunogen against PoRV infection as it can induce specific neutralizing antibodies, resulting in systemic immunity in piglets [5,6].

There is no effective treatment for PoRV infection, and vaccination is the primary method of control. Currently, PoRV commercial vaccines, such as attenuated [7] and subunit [8] vaccines, have been confirmed to protect against PoRV infection. However, their protection efficiency is limited. Therefore, there is an urgent need to develop a vaccine with a higher protection efficiency to prevent PoRV infection. Because PoRV mainly infects small intestinal epithelial cells, mucosal immunity plays a crucial role in protective immunity. ‘Live’ vaccines are the most effective vectors for inducing mucosal and systemic immunity [9,10]. At present, various microbes, such as *Salmonella*, *Escherichia coli*, and vaccinia virus, have been used to express and transmit foreign antigens in live vectors. Among these live vectors of oral vaccines, it was a major challenge that the antigen inserted into the live vector was potentially denatured and inactivated, leading to fever and diarrhea in the gut milieu [11,12,13].

Our previous studies suggest that lactic acid bacteria (LAB) are promising carriers for expressing and delivering exogenous antigens in oral vaccines [14,15,16]. Compared to other live vectors, LAB, which is a common bacteria in the intestinal tract, can adhere to and colonize the gastrointestinal tract [17]. LAB also produces a variety of bacteriocins with antibacterial and anti-diarrheal effects [18,19]. However, LAB must be safe as vaccine vectors, in addition to their probiotic properties. In order to achieve biosafety without antibiotic selection markers, LAB expression vectors with food-grade nutritional deficiency screening markers have been developed to attain biosafety and facilitate large-scale production. Food-grade selection markers can be divided into dominant and complementary markers. The cumbersome screening process for food-grade dominant markers limits their scope of application. Therefore, complementary markers have garnered considerable attention. The *alr* gene encoding alanine racemase food-grade selective marker has been widely used in biological genetic engineering [20,21]. The *alr* gene can catalyze the transformation of L-Alanine to D-Alanine, and D-Alanine is an important element in the bacterial cell wall biosynthesis of peptidoglycan. Because the *alr* gene is stably expressed in the genome of LAB without causing drug resistance, it can be used as a candidate marker for auxotrophy in food-grade LAB. Upon knockout of the genomic *alr* gene, *alr*-deficient LAB can grow normally only when D-Alanine is added externally or transferred into plasmids expressing *alr*-deficient complementary genes [22].

In this study, we constructed alanine racemase (*alr*) deficient LAB using the CRISPR-Cas9D10A gene editing system and an *alr* gene complementary expression plasmid to insert the gene of the major protective antigen VP4 of PoRV. Finally, we obtained a genetically engineered pPG-alr-VP4/*Δalr W56* strain that expressed the alr protein and VP4 protective antigens of PoRV. To provide a reference for developing an effective vaccine against PoRV, the immune effect of the recombinant strain was evaluated by oral immunization of the mice.

## 2. Materials and Methods

### 2.1. Bacterial Strain, Virus, Cell, Plasmid

The *Lactobacillus casei* strain W56 (isolated, identified and preserved by our laboratory [23] used in this study was cultured in de Man, Rogosa, and Sharpe (MRS) broth (Sigma, St. Louis, MO, USA) anaerobically without shaking at 37 °C. PoRV (A strain JL94, Genebank:AY523636) was isolated from clinical samples and purified by the plaque method prior to experimental work performed in a previous study [24]. Epithelial monkey kidney cells (MA104) were used to culture PoRV JL94 strain at 37 °C under 5% CO_2_. The *E. coli*-LAB shuttle expression plasmid pPG-T7g10-PPT contained the HCE constitutive secretory expression promoter, T7g10 translation enhancer, PgsA anchor protein, rrnBT1T2 transcription terminator, and other elements constructed in our laboratory and was used to construct the complementary plasmid pPG-alr-VP4 as described below. The gene-editing backbone plasmid pLCNICK was kindly supplied by Yang Sheng (Key Laboratory of Synthetic Biology, CAS, Shanghai, China).

### 2.2. Construction of Nutritionally-Deficient L. casei by CRISPR-Cas9D10A

According to the general principles of sgRNA design, the sgRNA sequences (Table 1) and the upstream/downstream homology arms (*alr*-HASup/HASdown) targeting *alr* sites were designed (Table 2), the two fragments were fused, and the fusion fragment named HASup-HASdown-sgRNA was introduced into the pLCNICK gene editing vector using *Xba*I and *Apa*I restriction endonuclease sites. The recombinant plasmid was named pLCNICK-alr (Figure 1A) and was electrotransformed into the *L. casei W56* strain to knock out the genomic *alr* gene. The recombinant strain was inoculated into an MRS plate medium containing 5 μg/mL erythromycin and 200 μg/mL D-Alanine, and a single colony was randomly selected to extract the genome. PCR was conducted to determine whether the alr gene was completely knocked out using the extracted *L. casei W56* genome as a template and *alr*-F/R as primers (Table 2). *L. casei W56*, with the *alr* gene deleted in the genome, was obtained through screening and identification and named *L. casei Δalr W56*.

### 2.3. Morphological Observation

Single colonies of *L. casei W56* and *L. casei Δalr W56* were inoculated into MRS liquid medium and cultured at 37 °C until the logarithmic growth phase. Then an appropriate amount of bacterial liquid was taken for gram staining. The bacterial liquid was evenly coated on the glass slide, dried and fixed, and then stained with ammonium oxalate crystal violet solution for 1 min. After washing, the bacterial cells were stained with gram iodine solution for 1 min and then decolorized with 95% alcohol solution for 30 s. Finally, the bacterial cells were counterstained with safranin for 30 s, washed with water, dried, and examined under an oil microscope to observe and compare the morphological characteristics.

The *L. casei W56* and *L. casei Δalr W56* strain was cultivated to a similar optical density value, and 4 mL of the culture was taken at OD_600_ of 0.5, 2.0 and 3.0, respectively. The bacterial pellet was collected, resuspended in 1 mL of ddH_2_O, and centrifuged at 3500 rpm for 5 min. Then the pellet was fully resuspended in pre-cooled 2.5% glutaraldehyde solution (pH = 7.2), and fixed overnight at 4 °C. After washing three times with 0.1 mol/L, pH = 7.2 phosphate buffer, the bacterial pellet was dehydrated by adding 50%, 70% and 90% ethanol solutions at 4 °C for 10 min. The bacterial pellet was harvested, dehydrated with 100% ethanol, and subjected to a displacement reaction. The obtained samples were first frozen and then dried and placed in a scanning electron microscope for observation after being coated.

### 2.4. Physiological and Biochemical Test

The viable counts of *L. casei W56* and *L. casei Δalr W56* were adjusted to about 1 × 10^8^ CFU/mL (OD_600_ = 1.0 ± 0.02) and inoculated into MRS liquid medium at a ratio of 1:100. The two strains were cultured at pH = 5.85 (normal medium), 2.0, 3.0 and 4.5, respectively, at 37 °C for 3 h. After doubling dilution with PBS, they were spread on MRS solid agar plates. The survival rate of the two strains was calculated after culturing at 37 °C for 24 h; 0.1% and 0.3% pig bile salts were added to *L. casei W56* and *L. casei Δalr W56* culture medium treated in the same way as above, and cultured at 37 °C for 3 h. After doubling the dilution with PBS, they were spread on MRS solid agar plates. The survival rate of the two strains was calculated after culturing at 37 °C for 24 h.

*L. casei W56* and *L. casei Δalr W56* single colonies were inoculated into different biochemical reaction tubes. After culturing at 32 °C for 10 h, the results were judged according to *Bergey’s Manual of Systermatic Bacteriology*.

### 2.5. Complementary Plasmid Generation

The *alr* gene was amplified with His-Tag, *Sac*I, and *Kpn*I restriction endonuclease sites, and the *VP4* gene was amplified with FLAG tags, *Kpn*I, and *Apa*I restriction endonuclease sites. Then the above two gene segments were co-connected to the *E. coli*-LAB shuttle vector pPG-T7g10-PPT to obtain the complementary plasmid pPG-alr-VP4 (Figure 1B). pPG-alr-VP4 and pPG-T7g10-PPT were electrotransformed into *L. casei Δalr W56*, and the recombinant LAB strains were named pPG-alr-VP4/*Δalr W56* and pPG/*Δalr W56*, respectively. PCR and sequence alignment was used to detect the stable presence of the alr-VP4 target gene in pPG-alr-VP4/*Δalr W56* by using the primers pPG-F/R (Table 2).

### 2.6. Western Blotting

To determine whether the alr-VP4 protein was expressed and assess its stability, the bacterial strains *L. casei Δalr W56*, pPG-alr-VP4/*Δalr W56*, and pPG/*Δalr W56* were passaged 50 times continuously in MRS broth, and the expression of the target protein was verified by Western blotting every ten generations. The proteins in the supernatant were harvested by lysis and subjected to sodium dodecyl sulfate-polyacrylamide gel electrophoresis after high-temperature denaturation. Proteins were then transferred to a polyvinylidene fluoride membrane after electrophoresis, and the target bands were detected using a mouse anti-His monoclonal antibody (Sigma, Ronkonkoma, NY, USA) at a dilution of 1:2000 as the primary antibody and HRP-conjugated goat anti-mouse IgG antibody at a dilution of 1:5000 (Sigma) as the secondary antibody. The expression of target protein bands was visualized using a chemiluminescence imaging system with ECL Plus Western blotting detection reagents (GE Healthcare, Chicago, IL, USA).

### 2.7. Animal Immunization

Four-week-old female BALB/c mice were obtained from Liaoning Changsheng Biotechnology Co., Ltd. (Shenyang, Liaoning Province, China). Strains *L. casei Δalr W56*, pPG-alr-VP4/*Δalr W56*, and pPG/*Δalr W56* were activated by streaking. Single colonies were picked and inoculated into MRS liquid medium with or without 10 µg/mL chloramphenicol and 200 µg/mL D-Alanine and cultured at 37 °C until the OD_600_ was approximately 1.0. The bacteria were washed once with PBS. After centrifugation, the bacteria were resuspended in PBS, and the concentration was adjusted to 10^10^ CFU/mL. The three groups mentioned above of bacterial liquid and PBS were orally inoculated into mice each time for three consecutive days, once every two weeks, for a total of three times. The number of mice and the immunization doses in the immunization program were shown in Table 3. Genital tract mucus, intestinal mucus, nasal fluid, feces, and blood samples were collected before immunization and on days 7, 14, 21, 28, 35, 42, 49, 56, and 63 after the initial immunization. Three parallel groups of each sample were sampled each time, and the samples from each group were analyzed as one sample in triplicate.

### 2.8. ELISA Assay

Genital tract mucus, intestinal mucus, nasal fluid, and fecal extracts were treated as previously described to detect SIgA antibodies [25]. In brief, all mucus samples were treated with 500 μL of PBS and mixed well (dilution not required). Feces were treated with 500 μL of EDTA-Na_2_-PBS (50 mmol/L), and the supernatant was collected after centrifugation (dilution not required). Mouse serum was diluted 10-fold to detect IgG antibodies and cytokine levels (IL-4, IL-2, IL-10, IL-12, IL-17, and IFN-γ). All samples were analyzed using the ELISA commercial kit (MEIMIAN, Enzyme industry Co., Ltd., Suzhou, Jiangsu, China).

Polystyrene microtiter plates were coated with PoRV as an antigen or PBS as a negative control and incubated overnight at 4 °C (0.1 mL per well). After blocking for two hours with 5% skim milk at 37 °C, the samples were added in triplicate and co-incubated with the antigen for one hour at 37 °C. HRP-conjugated goat anti-mouse IgG or IgA antibody at a dilution of 1:5000 (Sigma) was incubated for one hour at 37 °C. After washing, the color was developed using tetramethylbenzidine (Sigma), and the absorbance was measured at OD_450_.

### 2.9. Neutralization Assay

Mouse serum and intestinal mucus were collected on day 42 and sterilized by filtration to detect IgG and SIgA, respectively. After inactivation at 56 °C for 30 min, all samples were serially diluted two-fold to 1:256, and 0.1 mL of the diluted samples were added to 96-well cell culture plates. PoRV (0.1 mL of PoRV (10^2^ TCID_50_/mL)) was added to each culture well and co-incubated with the diluted samples for one hour at 37 °C. MA104 cells were seeded in 96-well culture plates and cultured at 37 °C for 18 h before infection. The cells were then infected with the coincubation mixture for two hours at 37 °C. The supernatant was removed, 0.1 mL of cell maintenance fluid was added to each well, and the TCID_50_ values were determined after incubation at 37 °C for 36 h. Neutralization titers were expressed as the reciprocal of the highest dilution of serum that showed at least a 50% reduction in the number of infected cells as compared with the negative control serum.

### 2.10. Statistical Analysis

Calculations were performed using the GraphPad Prism software (San Diego, CA, USA). All data are expressed as the mean ± standard deviation (x ± s). *p* values were calculated using the unpaired two-tailed Student’s *t*-test. *p* > 0.05 indicated statistical no significant difference (ns), *p* < 0.05 indicated statistical significance (*), *p* < 0.01 indicated high significance (**), *p* < 0.001 indicated extremely high significance (***).

## 3. Results

All animal experiments were approved by the Ethical Committee for Animal Experimentation of the Northeast Agricultural University, Harbin, China.

### 3.1. Construction of alr Nutrition-Deficient L. casei

The *alr* gene was knocked out in the *L. casei W56* genome using the CRISPR-Cas9D10A nickase-based plasmid pLCNICK-alr (Figure 1A) to obtain a nutritional deficient strain *L. casei Δalr W56*. The schematic diagram of the preparation of *L. casei Δalr W56* is seen in Figure 2A. After the *L. casei Δalr W56* strain was continuously transferred for 50 generations, the PCR results with *alr*-F/R primer (Table 1) showed that the target band (1626 bp) was smaller than that of the parental strain *L. casei W56* (2763 bp) (Figure 2B), and DNA sequencing confirmed that the *alr* gene was knocked out from the genome of *Δalr W56* (data not shown). These results indicated that the use of CRISPR-Cas9D10A nickase-based plasmid pLCNICK-alr could be capable of mediating rapid and efficient chromosomal deletions of the *alr* site.

The recombinant nutritionally deficient strain pPG-alr-VP4/*Δalr W56* was constructed via the electrotransformation of the complementary plasmid pPG-alr-VP4 (Figure 1B). We then tested the growth tendency of *L. casei Δalr W56*, pPG-alr-VP4/*Δalr W56*, and pPG/*Δalr W56* in MRS plates or MRS medium supplemented with or without D-Alanine. As shown in Figure 2C,D, *L. casei Δalr W56* and pPG/*Δalr W56* grew normally in MRS plates or MRS medium supplemented with D-Alanine but did not grow in MRS plates or MRS medium without D-Alanine. Moreover, pPG-alr-VP4/*Δalr W56* grew normally with or without D-Alanine in MRS plates or MRS medium. This indicated that *alr* was completely knocked out in *L. casei Δalr W56* and that *alr* was successfully expressed in pPG-alr-VP4/*Δalr W56*.

### 3.2. Biological Characterization of Δalr W56

In order to explore whether the deletion of *alr* gene affects the growth characteristics of the auxotrophic strain, we measured the growth curves of *L. casei W56* and *L. casei Δalr W56* under the condition of exogenous D-Alanine supplementation. The results showed that the growth curves of *L. casei Δalr W56* and *L. casei W56* strains were both S-shaped (Figure 3A). They both entered the logarithmic growth phase at 6 h; entered the stable phase at 18 h; then entered the decline phase. Compared with *L. casei W56*, the growth characteristics of the mutant strains *Δalr W56* did not change significantly.

The *L. casei Δalr W56* and *L. casei W56* strains in the logarithmic growth phase were subjected to gram staining, and the morphological characteristics of the bacteria were observed under the microscope. The results showed that the mutant strain was blue-purple, short rod-shaped, and in a chain-like arrangement, which was consistent with *L. casei W56* (Figure 3B). Similarly, the scanning electron microscope results also showed that compared with *L. casei W56*, the morphological characteristics of *L. casei Δalr W56* did not change significantly (Figure 3C).

Acid and bile salt resistance experiments were carried out on *L. casei Δalr W56* and *L. casei W56* strains. The results showed that both *L. casei Δalr W56* and *L. casei W56* strains were resistant to 0.1% bile salts and could grow slowly in 0.3% bile salts concentration (Table 4). In addition, both strains were resistant to the acidic environment of pH = 2, 3, or 4.5, but the number of viable bacteria decreased as the pH value decreased (Table 4). The above results showed that there was no significant difference in acid and bile salt resistance between *L. casei Δalr W56* and *L. casei W56* strains and all of them could survive in an environment that simulated gastrointestinal digestive juices in vitro.

The biochemical reaction results of *L. casei Δalr W56* and *L. casei W56* strains are shown in Table 5. The results showed that the nitrate reduction test, catalase test, indole test and gelatin liquefaction test of the two strains were negative. Combined with the gram staining results, it was proved that both *L. casei Δalr W56* and *L. casei W56* strains conformed to the biochemical characteristics of *Lactobacillus*. In addition, other biochemical reactions of the L. *casei Δalr W56* strain were basically consistent with the *Lactobacillus casei* in the *Bergey’s Manual of Systermatic Bacteriology*, indicating that the biochemical characteristics of the *L. casei Δalr W56* strain did not change significantly. Together, these results demonstrate that the deletion of the *alr* gene does not affect the biological characteristics of the *L. casei W56* strains.

### 3.3. Inherited Stability of the Δalr W56 Expressing alr-VP4

To confirm that the target protein was successfully expressed in the pPG-alr-VP4/*Δalr W56* strain, the cell lysates of pPG-alr-VP4/*Δalr W56*, *L. casei Δalr W56*, and pPG/*Δalr W56* were analyzed using Western blotting. As shown in Figure 4A, the alr-VP4 protein (70 kDa) was detected in pPG-alr-VP4/*Δalr W56* but not in *L. casei Δalr W56* or pPG/*Δalr W56*.

The recombinant strain (pPG-alr-VP4/*Δalr W56*) was serially inoculated 50 times to evaluate its stability. The 10th, 20th, 30th, 40th, and 50th generations of recombinant strain plasmids were extracted, and the proteins were lysed. The inherited stability was tested using PCR and Western blotting. All recombinant strains amplified the alr-VP4 sequence 1968 bp (Figure 4B), and the expression of the alr-VP4 protein (70 kDa) was observed (Figure 4C). These results indicated that recombinant *L. casei* pPG-alr-VP4/*Δalr W56* successfully expressed alr-VP4 and had good inherent stability.

### 3.4. Antibody Responses Post Oral Immunization

To analyze the immunogenicity of the recombinant strain (pPG-alr-VP4/*Δalr W56*), we collected samples from the mice at different time points after oral immunization. A schematic of the oral immunization program and sampling schedule is shown in Table 4 and Figure 5A. The ability of pPG-alr-VP4/*Δalr W56* to elicit mucosal and systemic immunity was evaluated by determining the specific IgG and SIgA antibodies, respectively. No significant changes were observed in serum levels of anti-PoRV-specific IgG and SIgA antibodies before immunization. The results demonstrated that the SIgA antibody levels in the genital tract fluid (Figure 5B), intestinal mucus (Figure 5C), nasal fluid (Figure 5D), and feces (Figure 5E) of mice treated with pPG-alr-VP4/*Δalr W56* increased on the seventh day after the initial immunization and peaked on the 42nd day. After booster immunization, significantly higher SIgA titers were detected in these mice than in those treated with PBS, *L. casei Δalr W56*, or pPG/*Δalr W56*.

Furthermore, anti-PoRV IgG antibody levels in mouse serum increased on the seventh day after the first oral immunization with pPG-alr-VP4/*Δalr W56* and were significantly higher than those in mice treated with PBS, *L. casei Δalr W56*, and pPG/*Δalr W56* (Figure 5F). The IgG antibody titers peaked on the 14th day after the third immunization (42nd day after the initial immunization) and were significantly higher than those in PBS, *L. casei Δalr W56*, and pPG/*Δalr W56*. These results suggest that the recombinant strain (pPG-alr-VP4/*Δalr W56*) expressing the PoRV-VP4 antigen effectively elicited an anti-PoRV local and systemic immune response in vivo.

### 3.5. In Vitro PoRV-Neutralizing Activity of Antibodies

Based on the detection results of specific IgG and SIgA antibody titers against PoRV collected at different time points, we detected the neutralizing activity of the serum IgG antibody and intestinal mucosa SIgA antibody on day 42 after the initial immunization. As shown in Figure 5G, the neutralizing activity of the serum IgG antibody against PoRV in mice immunized with pPG-alr-VP4/*Δalr W56* was 1:51.4, which was more robust than that in mice orally administered PBS (1:2), *L. casei Δalr W56* (1:2) and pPG/*Δalr W56* (1:2). The intestinal mucosa SIgA antibody in mice that received pPG-alr-VP4/*Δalr W56* also possessed stronger anti-PoRV neutralizing activity (1:22.9) than in mice orally administered PBS (1:2), *L. casei Δalr W56* (1:2), and pPG/*Δalr W56* (1:2). This result proves that the recombinant strain pPG-alr-VP4/*Δalr W56* can induce effective neutralizing antibodies in mice under the immunization program in this study, and reveals that the auxotrophic Lactobacillus *Δalr W56* are feasibile as antigen delivery carriers to construct oral immunization vaccines.

### 3.6. Cytokine Secretion Level in Immunized Mice

We tested the serum cytokines produced by pPG-alr-VP4/*Δalr W56* in mice after oral immunization and found that higher levels of the Th1-associated cytokine IFN-γ and Th2-associated cytokine IL-4 were induced compared to those in the control groups (PBS, *L. casei Δalr W56*, and pPG/*Δalr W56*; *p* < 0.01) (Figure 6A). In order to determine the type of cellular immunity induced by pPG-alr-VP4/*Δalr W56* in mice, the IL-4/IFN-γ ratio was calculated and found to be greater than 1, and significantly higher than the other control groups (Figure 6B). Moreover, a tendency similar to that of IFN-γ and IL-4 was observed for IL-2, IL-10, IL-12, and IL-17 (Figure 6C). These results indicate that pPG-alr-VP4/*Δalr W56* can significantly stimulate Th1, Th2, and Th17 cell immunity in mice. Furthermore, the IL-4/IFN-γ ratio was greater than 1, indicating that the recombinant strain mainly induced the Th2 type humoral immune response.

## 4. Discussion

PoRVs are the primary cause of acute diarrhea in piglets, leading to substantial economic losses in pig farming. However, there is currently no effective treatment for the inhibition of PoRV infection [24]. Vaccination is the most effective method for preventing PoRV infection in piglets. Antigens are pivotal elements of vaccine development (1). Previous studies have shown that VP4 of PoRV can induce neutralizing antibodies to protect herds from PoRV infection [26,27]. Thus, VP4 is an ideal antigen for developing an anti-PoRV vaccine. In addition to antigens, vectors are an essential factor in vaccine development. As PoRV replication occurs in the epithelial cells of the intestinal villi [28,29], the most effective method to prevent PoRV infection is mucosal immunity in the intestinal tract induced by an oral vaccine. Several studies have reported that *L. casei* is an ideal candidate for use in oral vaccines [26,28,30]. Qiao et al. [22] constructed a recombinant *L. casei* strain expressing the PoRV VP4 gene and LTB mucosal adjuvant. Oral immunization of mice significantly induced the production of serum IgG and mucosal SIgA antibodies, which had a significant neutralization effect against PoRV infection. Yin et al. constructed recombinant LAB without antibiotic selection markers and constitutively stably expressed the PoRV-neutralizing epitope VP4 protein, which immunized mice and also showed a good immune protection effect [31]. In summary, as an intestinal probiotic, *L. casei* can maintain the balance of intestinal flora and promote the digestion and absorption of intestinal nutrients as well as enhance immune regulation [28,30,32,33,34,35,36].

Overcoming drug resistance caused by resistance genes and manual control of the screening of recombinant strains were crucial for selecting *L. casei* as the vector for the oral vaccine in this study. Developing a food-grade selection marker without antibiotic resistance genes in food-grade microorganisms is vital for constructing genetically modified LAB products that meet biosafety standards. The construction of a food-grade nutritionally-deficient screening marker has an effective application in a non-resistance screening system [37]. However, the use of traditional plasmid targeting systems to manipulate the genome of LAB has raised numerous concerns, such as the difficulty of knocking out or inserting foreign genes, low expression efficiency and reorganization efficiency, and residues of transposon plasmids and other element sequences. CRISPR-Cas9 technology relies on Cas9 and sgRNA to achieve precise cutting, simplify the experimental steps, and reduce the experimental costs. Genome-editing tools mediated by the CRISPR-Cas9 system have been used in Lactobacillus bacteria [38,39,40,41,42]. Oh et al. [43] used Cas9 protein to knock out unedited cells of wild strain, which significantly improved the genome editing efficiency of ssDNA for *Lactobacillus reuteri*. Guo et al. [44] combined ssDNA with a modified CRISPR-Cas9 counter-selection to successfully knock out the Lactococcus lactis UPP gene with a genome editing efficiency of 75%. However, Cas9 can cause dsDNA breakage (DSB) in the bacterial genome, leading to bacterial death [45,46]. To overcome this, Song et al. [41] used the Cas9D10A-Nickase mutant to replace the wild-type Cas9 protein and optimized the sgRNA and Cas9D10A promoter, which not only achieved precise knockout of gene segments and insertion of foreign genes into the genome of *L. casei* but also demonstrated that the Cas9D10A-Nickase system is an effective tool to overcome DSB-induced bacterial lethality.

In this study, we used the CRISPR-Cas9D10A gene editing system to cut the single-stranded genomic DNA of *L. casei W56*, thereby knocking out the *alr* gene, and the auxotrophic strain *L. casei Δalr W56* can be stably passaged without mutation reversion. Simultaneously, we constructed a LAB expression plasmid, pPG-alr-VP4, complementary to *L. casei Δalr W56*. We verified that *L. casei Δalr W56* could resume normal growth by exogenous supplementation with the *alr*-missing gene, indicating that the host bacteria and plasmids achieve functional complementarity. We also explored the expression of the PoRV protective antigen VP4 using the complementary plasmid. The results showed that the recombinant *L. casei* pPG-alr-VP4/*Δalr W56* expressing VP4 protein can induce mice to produce effective mucosal and systemic immune responses against PoRV. This indicates that this gene editing system can effectively carry out precise editing of the *Lactobacillus* genome, and the complementary plasmids can also achieve the immune function of recombinant strains by expressing foreign antigens.

As PoRV mainly infects small intestinal epithelial cells after passing through the mucosal layer, the mucosal immune system is important in the prevention and control of PoRV infection. Our results indicated that oral administration of pPG-alr-VP4/*Δalr W56* can effectively induce specific serum IgG and mucosal SIgA antibodies and have neutralizing activity against PoRV. The SIgA-mediated protection of mucosal immunity in small intestinal epithelial cells plays an essential role in preventing PoRV infection [28]. Serum IgG antibodies also play pivotal roles in host immune defense and enhance humoral immunity [47]. Generally speaking, the immune response after vaccination has a response period. IgG appears before IgA and induces a longer duration than the IgA. Moreover, SIgA as a secreted antibody, binds to antigenic microorganisms to form a complex, thereby activating complement to exerting immune function. Therefore, when an antigen is bound, part of SIgA is also consumed [48,49,50]. In this study, the booster immunization occurred 2 weeks after the primary immunization, and at this time the concentration of IgG remained a high level in the serum of the mice, but the concentration of SIgA decreased due to the immunization time interval and functional consumption. After booster immunization again, the SIgA antibodies levels increased significantly and peaked on the 42nd day. This result also suggests that our immunization program can be further optimized. For the animal model in this study, increasing the oral dose within a reasonable range and shortening the immunization interval may enable the recombinant oral vaccine to stimulate the body to produce high-efficiency SIgA faster and exert a better immune response effect.

In this study, after oral immunization in mice, the pPG-alr-VP4/*Δalr W56* strain could colonize the intestine, and then stimulate the intestinal mucosal immune system to produce specific SIgA. When the sensitized lymphocytes of the local mucosa enter the blood circulation, they will gradually differentiate and mature. Under the mediation of different receptors, they can reach other mucosal sites and induce the special SIgA for activation of the immune system, this phenomenon is known as the common mucosal immune system (CMIS) [51,52]. Thus, although the oral recombinant vaccine mainly stimulates the intestinal mucosal immune system, the specific SIgA can be detected in nasal fluid, genital tracts and feces. Furthermore, PoRV-specific IgG antibodies were detected in the serum, and the dynamic changes were consistent with SIgA in intestinal mucus. It has been proved by in vitro experiments that they can effectively neutralize PoRV. PoRV as a porcine enterotropic virus mainly infects epithelial cells to impair the function of the digestive tract. Multiple enzymes in the digestive tract may destroy the complete IgG structure and affect its ability to bind antigens. SIgA is protected by SC fragments and the J-chain (joining chain) to avoid hydrolysis by digestive enzymes [53]. Therefore, SIgA plays a more important role than IgG in protecting piglet intestinal epithelial cells from PoRV infection. These results illustrate that the oral administration of pPG-alr-VP4/*Δalr W56* can provide effective immunity against PoRV infection by inducing high levels of mucosal SIgA and serum IgG antibodies.

When stimulated by foreign antigens, the body responds with Th1-type cellular immune responses and Th2-type humoral immune responses in mammals. The degree of immunogenicity that induces an effective immune response in the body is an important indicator for estimating the efficacy of an oral vaccine. In this study, to comprehensively evaluate the immune response induced by pPG-alr-VP4/*Δalr W56*, cytokine levels were determined after the oral administration of the vaccine. IFN-γ and IL-2 are Th1-associated cytokines, which contribute to antibody production and participate in the cellular immune response, T lymphocyte proliferation and induce cytotoxicity [54]. IL-4 and IL-10 produced by Th2 helper T cells mainly promote B cell activation, proliferation, differentiation, and antibody production, and plays an important role in regulating humoral immune responses [55]. In addition, IL-6 also produced by Th2-type cells can induce the differentiation of Th17 cells, thereby producing IL-17. IL-17 can prevent the spread of pathogens through the intestinal route and enhance the protective effect of the intestinal mucosal barrier [56]. In this study, the recombinant strain expressing PoRV neutralizing epitope VP4 protein induced the immune response dominated by Th2, which plays an important role in the body’s antiviral response.

## 5. Conclusions

In this study, we knocked out the *alr* gene in the genome of *L. casei* using the CRISPR-Cas9D10A gene editing system and constructed a complementary plasmid expressing alr-VP4 simultaneously. Construction of the recombinant strain (pPG-alr-VP4/*Δalr W56*) used food-grade selective markers instead of traditional resistance gene markers to realize the artificial operation of the strain and has completely independent intellectual property rights. We subsequently demonstrated that the recombinant strain could efficiently induce specific local mucosal immunity, systemic humoral immunity, and cellular immunity via oral immunization in mice, suggesting that it is a promising strategy for developing an oral vaccine against PoRV infection.

## Figures and Tables

**Figure 1 vaccines-10-01510-f001:**
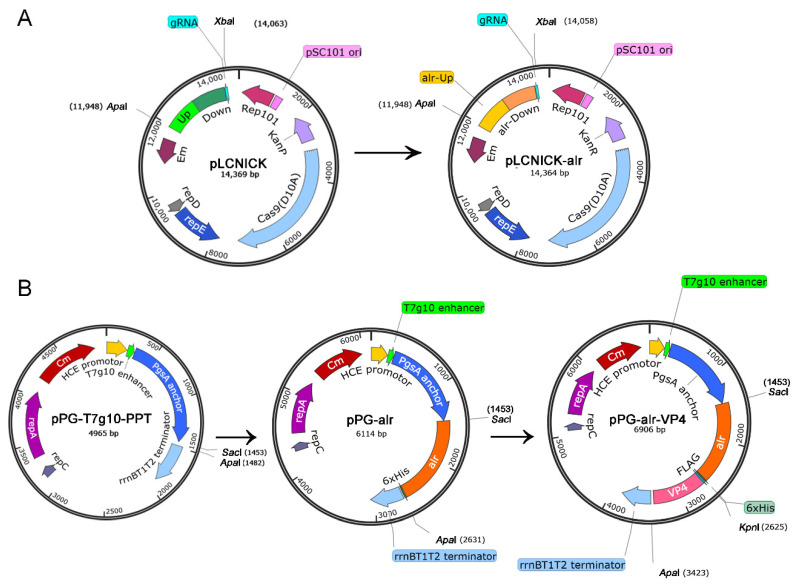
Schematic diagram of the construction of DNA plasmids. (**A**) The sgRNAs of *alr* were fused with the upper and lower homologous arms of the alr gene, and the sequence was inserted into the pLCNICK gene editing vector, which resulted in the pLCNICK-alr gene editing vector. (**B**) The constitutive cell surface expression plasmid pPG-T7g10-PPT (**left**). The alr gene was amplified from *L. casei W56*, cleaved using *Sac*I and *Apa*I, and was inserted into the expression plasmid pPG-T7g10-PPT, named pPG-alr (**middle**). The VP4 gene was amplified from the PoRV JL94 strain, cleaved with *Kpn*I and *Apa*I, and was inserted into pPG-alr, obtaining the recombinant plasmid pPG-alr-VP4 (**right**).

**Figure 2 vaccines-10-01510-f002:**
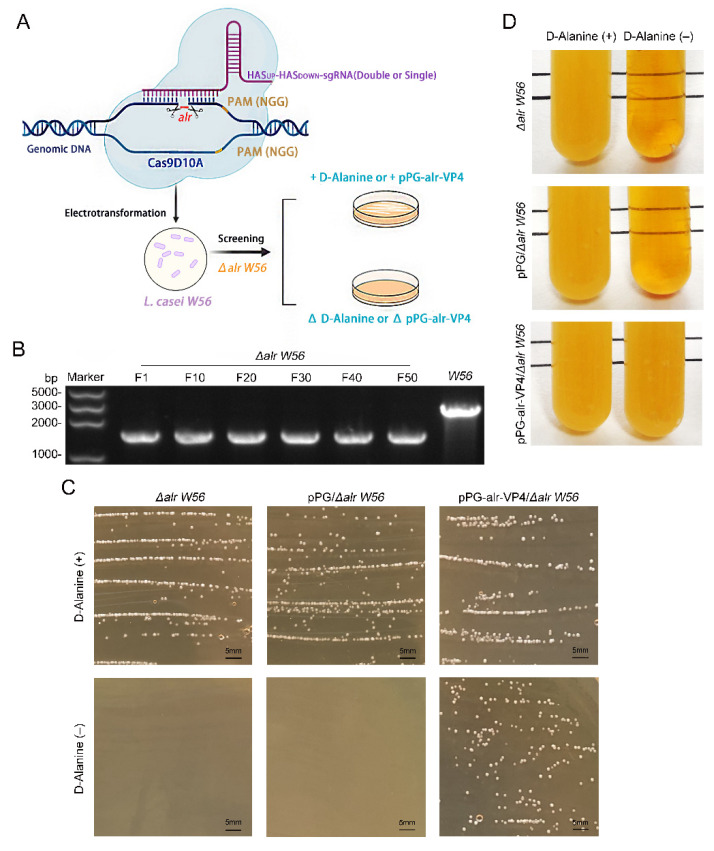
Construction of *alr*-auxotrophic mutant *L. casei Δalr W56*. (**A**) Schematic diagram of the preparation of *L. casei Δalr W56* using the CRISPR/Cas9D10A system. (**B**) Identification of *alr* gene depletion in the genomes of *L. casei W56* and *L. casei Δalr W56* by PCR using *alr*-F/R as primers. *L. casei Δalr W56* was serially passaged 50 times and the genome extracted every 10 passages was identified by PCR and DNA sequencing. (**C**) Strains *L. casei Δalr W56*, pPG/*Δalr W56*, and pPG-alr-VP4/*Δalr W56* were cultivated on MRS solid plates with and without D-Alanine. (**D**) Strains *L. casei Δalr W56*, pPG/*Δalr W56*, and pPG-alr-VP4/*Δalr W56* were cultivated in MRS liquid medium with and without D-Alanine.

**Figure 3 vaccines-10-01510-f003:**
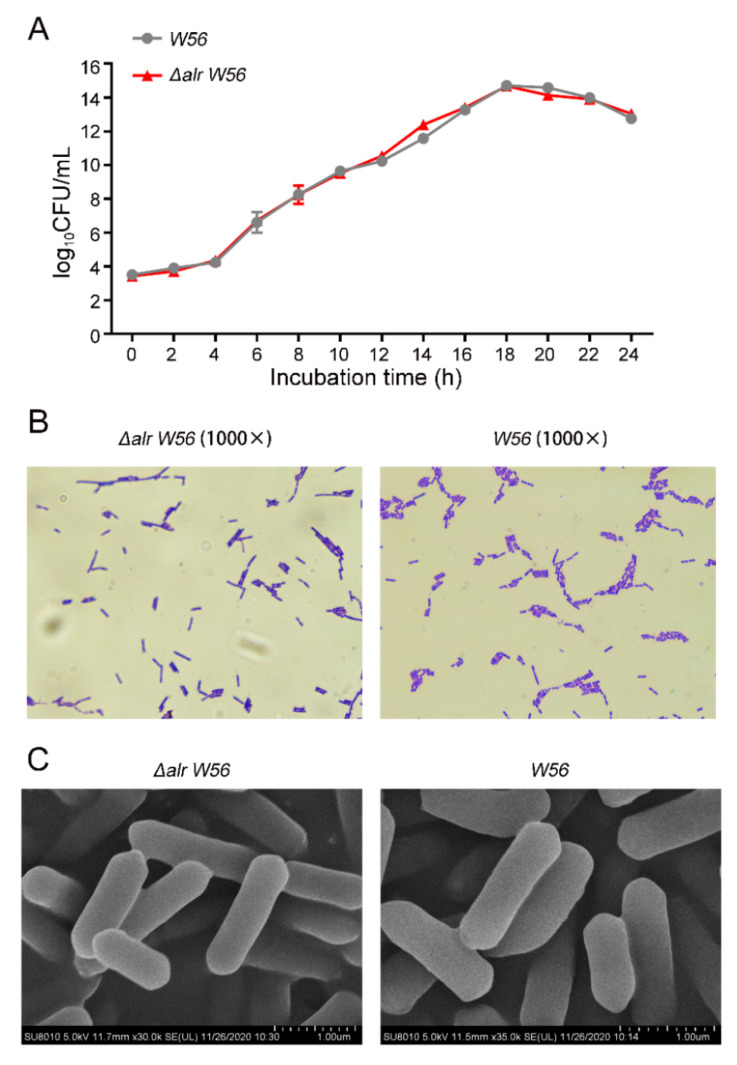
Biological characterization of L. casei Δalr W56. (**A**) Determination of growth curves of *L. casei W56* and *L. casei Δalr W56*. Colony Forming Units (CFU) per milliliter of *L. casei W56* and *L. casei Δalr W56* cultures were calculated by Standard Plate Count (SPC) every 2 h respectively. (**B**) The morphological characteristics of *L. casei W56* and *L. casei Δalr W56* were identified by gram staining under the ordinary light microscope, and the magnification of view was 1000×. (**C**) Morphological characteristics of *L. casei W56* and *L. casei Δalr W56* observed by scanning electron microscope.

**Figure 4 vaccines-10-01510-f004:**
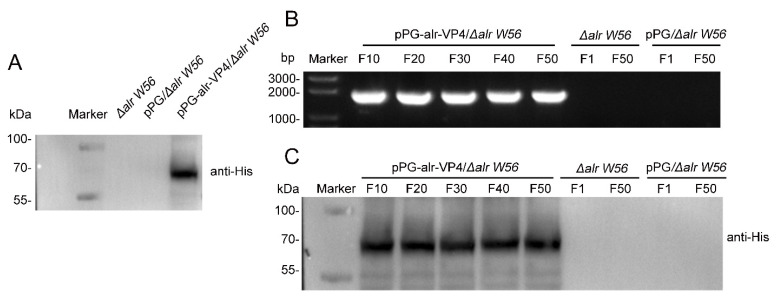
Analysis of the inherent stability in recombinant strains expressing alr-VP4. (**A**) Strains pPG-alr-VP4/*Δalr W56*, *L. casei Δalr W56*, and pPG/*Δalr W56* were lysed and the protein expression levels of alr-VP4 were analyzed using Western blotting with mouse anti-His monoclonal antibody. (**B**) Strains pPG-alr-VP4/*Δalr W56*, pPG/*Δalr W56*, and *L. casei Δalr W56* were continuously transferred for 50 generations and were lysed every 10 (pPG-alr-VP4/*Δalr W56*) or 50 (*L. casei Δalr W56* and pPG/*Δalr W56*) generations. The genome was extracted, and alr-VP4 was amplified by PCR. (**C**) Strains pPG-alr-VP4/*Δalr W56*, pPG/*Δalr W56*, and *L. casei Δalr W56* were continuously transferred for 50 generations and were lysed every 10 (pPG-alr-VP4/*Δalr W56*) or 50 (pPG/*Δalr W56* and *L. casei Δalr W56*) generations, and the protein expression levels of alr-VP4 were analyzed using Western blotting with mouse anti-His monoclonal antibody.

**Figure 5 vaccines-10-01510-f005:**
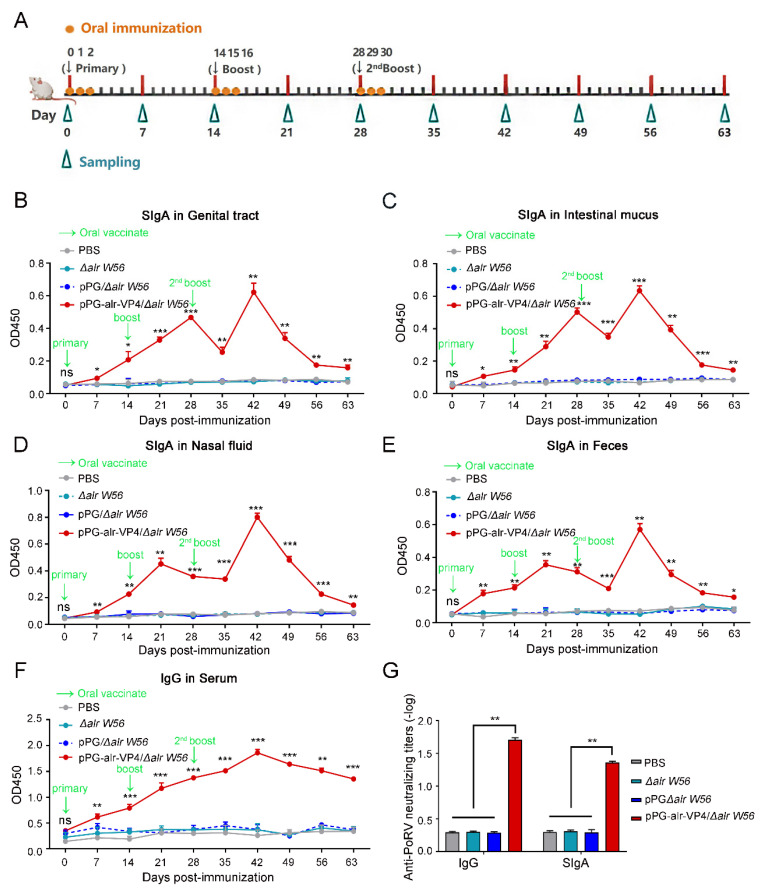
Levels of anti-PoRV-specific SIgA and IgG in immunized mice. (**A**) Schematic of oral immunization program and schedule of sampling genital tract fluid, intestinal mucus, nasal fluid, feces, and sera. The immunization program was immunization each time for three consecutive days, once every two weeks, for a total of three times. Samples were taken every 7 days until day 63. (**B**–**E**) Specific anti-PoRV SIgA levels in genital tract (**B**), intestinal mucus (**C**), nasal fluid (**D**), and feces (**E**) of mice were determined post-immunization with strains *L. casei Δalr W56*, pPG/*Δalr W56*, pPG-alr-VP4/*Δalr W56*, or PBS. (**F**) Detecting the level of anti-PoRV specific IgG antibody in serum. (**G**) The neutralizing activity of anti-PoRV specific IgG antibody in serum and SIgA in intestinal mucosa were detected. Three parallel groups of each sample were sampled each time, and the samples from each group were analyzed as one sample in triplicate. n = 3, mean ± SD, Student’s test, ns, not significant, *, *p* < 0.05, **, *p* < 0.01, ***, *p* < 0.001.

**Figure 6 vaccines-10-01510-f006:**
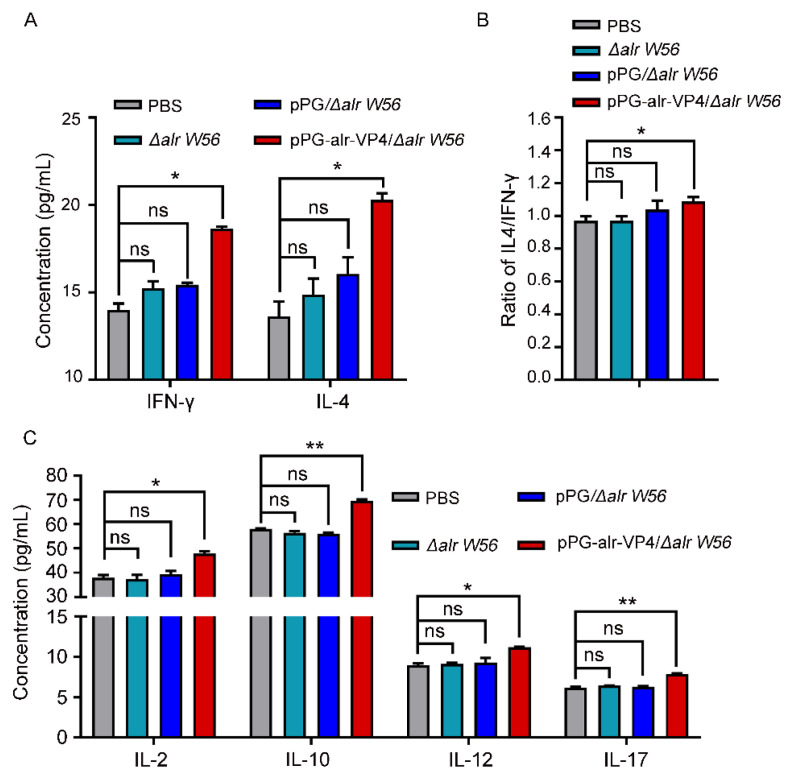
Cytokine secretion level in immunized mice. The mice were immunized thrice with strains *L. casei Δalr W56*, pPG/*Δalr W56*, pPG-alr-VP4/*Δalr W56* or PBS. (**A**) The serum was collected and the cytokine for IFN-γ or IL-4 was detected. (**B**) The ratio of IL4/IFN-γ was analyzed. (**C**) The serum was collected and the cytokines for IL-2, IL-10, IL-12, and IL-17 were detected. n = 3, mean ± SD, Student’s test, ns, not significant, *, *p* < 0.05, **, *p* < 0.01.

**Table 1 vaccines-10-01510-t001:** Sequence of sgRNAs targeting *alr* gene.

ID	sgRNAs Sequence (5′-3′)
Double-sgRNA-1	GCAGCTTTTCGCTGTGGTCA
Double-sgRNA-2	GGTTGGGCAAAGTGGCGATC
Single-sgRNA	GCAGCTTTTCGCTGTGGTCA

**Table 2 vaccines-10-01510-t002:** Details of primers.

ID	Primer Sequence (5′-3′)	Target
*alr*-HAS_up_-F/R	F: tctttttctaaactagggccc ^1^ GAATCGCGCCACTGGCCA R: aaagaaggtcctg ^1^ TGATGGTGGCTCGTATGCCC	*alr*-upstream homology arms
*alr*-HAS_down_-F/R	F: acaccatca ^1^ CAGGACCTTCTTTTTCTAAAATTACCT R: agtcggtgctttttttgag ^1^ CCAAACGAAATCGAATATTTGCA	*alr*-downstream homology arms
*alr*-F/R	F: GACCAGACCCACTGAAATCG R: AACCACCAACAGCAGAAGAA	*alr*
pPG-F/R	F: GACAGCCTTAAACAGAAAACC R: GCAGTTCCCTACTCTCGC	alr-VP4

^1^ Lowercase letters are homologous binding sequences.

**Table 3 vaccines-10-01510-t003:** The groups and dosage of immunization.

Groups	Dosage	Number of Mice
pPG-alr-VP4/*Δalr W56*	10^10^ CFU	30
pPG/*Δalr W56*	10^10^ CFU	30
*L. casei Δalr W56*	10^10^ CFU	30
PBS	100 µL	30

**Table 4 vaccines-10-01510-t004:** Acid and bile salt resistance test.

Strains ID	Viability (%)				
	0.1% Bile Salt	0.3% Bile Salt	pH2	pH3	pH4.5
*L. casei Δalr W56*	1.82 ± 0.51	0.42 ± 0.04	0.76 ± 0.12	5.05 ± 1.55	50.12 ± 1.4
*L. casei W56*	1.88 ± 0.41	0.46 ± 0.19	0.83 ± 0.17	5.78 ± 1.18	53.52 ± 2.35

**Table 5 vaccines-10-01510-t005:** The results of biochemical test.

Reaction Type	Nitrate Reduction	Catalase Test	H_2_S	Indole Test	Gelatin Liquefaction	Glucose Fermentation	Melezitose	Lactose	Maltose	Raffinose	Mannose	Salicin	Mannitol	Melibiose	Rhamnose	Ribose	Sorbose	Sucrose	Xylose	Mushroompolysaccharide	Esculoside	Fructose	Arabinose	Galactose	Cellobiose	Citrate	Malonate	Gluconate	Arginine Dihydrolase	Arginine	Amygdalin
BIM1	**−**	**−**	**−**	**−**	**−**	**+**	**+**	**−**	**+**	**−**	**+**	**+**	**+**	**−**	**−**	**+**	**+**	**+**	**−**	**+**	**+**	**+**	**−**	**+**	**+**	**−**	**−**	**+**	**−**	**−**	**+**
*L. casei*	**−**	**−**	**−**	**−**	**−**	**+**	**−**	**−**	**+**	**−**	**+**	**+**	**+**	**−**	**−**	**+**	**+**	**+**	**−**	**+**	**+**	**+**	**−**	**+**	**+**	**−**	**−**	**+**	**−**	**−**	**+**
*Δalr W56*	**−**	**−**	**−**	**−**	**−**	**+**	**−**	**−**	**+**	**−**	**+**	**+**	**−**	**−**	**−**	**+**	**+**	**+**	**−**	**+**	**+**	**+**	**−**	**+**	**+**	**−**	**−**	**+**	**−**	**−**	**+**

## Data Availability

Not applicable.

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
