# Peer review of "Auxotrophic Lactobacillus Expressing Porcine Rotavirus VP4 Constructed Using CRISPR-Cas9D10A System Induces Effective Immunity in Mice"

_vaccines, 2022, doi:10.3390/vaccines10091510_

Round 1

Reviewer 1 Report

Zhang et al knocked out the alanine racemase (alr) gene in the genome of L. casei W56 to construct an auxotrophic L. casei strain (L. casei Δalr W56) using the CRISPR-Cas9D10A gene editing system and made a recombinant strain (pPG-alr-VP4/Δalr W56) to express alr protein and VP4 protective antigens of PoRV. They have used the recombinant strain in various preliminary experiments to characterize, evaluate and propose it as a safer vaccine strategy against PoRV infection. 

Minor points:

Page 2, line 60, strain names are not in italics.

Page 2, line 94, ref 56 has been cited but not listed in the reference section. 

Similarly, the strain’s names in the figure captions were not italicized.

Page 3, line 99, strain name E. coli is not in italics.

Page 5, line 164, check the last sentence.

Page 5, link 186, check the last word in the last para, ‘Animal immunization’ – was it the heading of the next section?

In table 5, the first row is not readable, there are some formatting issues.

Author Response

Response to reviewer: Thank you very much for your comments.

We have italicized the strain names on lines 60, 99 and all figures, and added reference 56 (Now changed to 23) in line 94.

Based on your suggestion, we have adjusted the word order on line 164 for easier understanding.

Additionally, we corrected the sequence number of the heading ‘Animal immunization’ on line 186.

We've reformatted table5 to make it readable.

Reviewer 2 Report

In the manuscript titled “Auxotrophic Lactobacillus Expressing Porcine Rotavirus VP4 Constructed Using CRISPR-Cas9D10A System Induces Effective Immunity in Mice”, these authors constructed alanine racemase (alr) deficient LAB using the CRISPR-Cas9D10A gene editing system and an alr gene complementary expression plasmid to insert the gene of the major protective antigen VP4 of PoRV. Finally, they obtained a genetically engineered pPG-alr-VP4/Δalr W56 strain that expressed the alr protein and VP4 protective antigens of PoRV. To provide a reference for developing an effective vaccine against PoRV, the immune effect of the recombinant strain was evaluated by oral immunization of the mice. The manuscript is well written and easy to follow. The results are presented very well followed by a nice discussion. In my view, the manuscript is suitable for publication in this journal.

Major comment:

It would be perfect if the authors use the deficient LAB expressing VP4 to protect individuals from infection of PoRV in animal model. As readers, we are curious about the efficiency. In figure 5G, the authors showed in vitro PoRV-neutralizing activity of antibodies, if possible, they need performed the in vivo experiment.

Manor comment:

Text in table 5 is not so clear.

Author Response

Response: Thank you for your comments. We value reviewer's constructive suggestions with great appreciation. In fact, we have tried to infect mice with the isolated PoRV strains. However, we found that after artificial infection of mice, the isolated porcine rotavirus did not show obvious clinical features like natural disease, and it is more difficult to observe the phenomenon of mice death. We think that the death caused by rotavirus infection is mostly caused by water salt imbalance after osmotic diarrhea. Another reason is that rotavirus is often infected mixing with other pathogens or secondary infection. Infection of rotavirus is generally a selective incentive of animals' death, only PoRV-infected mice may not produce typical clinical features. Therefore, it is difficult to objectively evaluate the immune protection against PoRV in mice. So, the protective test in mice of challenge was not tested. But we also think this research is critical for us to develop an effective vaccine against PoRV. In the next stage of research, by establishing a PoRV infection model in pigs, we will further comprehensively and systematically evaluate the immune effect of the recombinant L.casei vaccine in pigs and optimize immunization procedures, as well as evaluate host immunity protection induced by L.casei vaccine to develop effective oral vaccines for controlling PoRV infection.

Reviewer 3 Report

The authors aimed to construct alanine racemase (alr) deficient LAB using the CRISPR-Cas9D10A gene editing system and a gene complementary expression plasmid (alr) to insert the gene of the major protective antigen VP4 of PoRV.

The data were important, because the outcomes showed a genetically engineered pPG-alr-VP4/Δalr W56 strain that expressed the alr protein and VP4 protective antigens of PoRV. This in the view to provide a reference for developing an effective vaccine against PoRV, the immune effect of the recombinant strain was evaluated by oral immunization of the mice. Therefore, I have no further comments against the manuscript, but only one suggestion: Limitations of the study needs to be added and discussed.

Author Response

Response: Thank you for your comments. We value Reviewer's constructive suggestions with great appreciation. We have added a discussion of the immunization program in this study and added feasibility options in lines 469-473 of the Discussion section.